# “Not Only Adults Can Make Good Decisions, We as Children Can Do That as Well” Evaluating the Process of the Youth-Led Participatory Action Research ‘Kids in Action’

**DOI:** 10.3390/ijerph17020625

**Published:** 2020-01-18

**Authors:** Manou Anselma, Mai Chinapaw, Teatske Altenburg

**Affiliations:** Department of Public and Occupational Health, Amsterdam Public Health research institute, Amsterdam UMC, Vrije Universiteit Amsterdam, Van der Boechorststraat 7, NL-1081 BT Amsterdam, The Netherlands; m.chinapaw@amsterdamumc.nl (M.C.); t.altenburg@amsterdamumc.nl (T.A.)

**Keywords:** children, participation, health behavior, low-income

## Abstract

In Youth-led Participatory Action Research (YPAR), youth collaborate with academic researchers to study a problem, develop actions that align with their needs and interests, and become empowered. ‘Kids in Action’ aimed to develop actions targeting healthy physical activity and dietary behavior among, and together with, 9–12-year-old children as co-researchers. This paper presents the process evaluation of ‘Kids in Action’ based on eight focus groups with children (*N* = 40) and eight interviews with community partners (*N* = 11). Interview guides were based on empowerment theory and the RE-AIM framework, in order to evaluate the study on: empowerment, collaborations, reach, effectiveness, adoption, implementation, and maintenance. Transcripts were analyzed using evaluation and provisional coding. Both children and community partners perceived an increased awareness of healthy behaviors and an improvement in confidence, critical awareness, leadership and collaboration skills, which contributed to increased feelings of empowerment. Community partners valued child participation and the co-created actions. Actions were also well-perceived by children and they liked being involved in action development. The strong relationship of researchers with both children and relevant community partners proved an important facilitator of co-creation. Future studies are recommended to attempt closer collaboration with schools and parents to gain even more support for co-created actions and increase their effectiveness.

## 1. Introduction

Youth-led Participatory Action Research (YPAR) is a methodology in which academic researchers collaborate with youth to make improvements in their. Youth who participate in YPAR identify issues in their community that they want to improve, and through conducting research, find starting points to take action and make change happen [1,2]. They become change agents and build the power to improve their communities [3]. Youth are seldom involved as co-researchers [4], but as experts of their own lives, they can provide valuable insights into their community, lives, and behaviors [5]. In YPAR, youth participate in the research process as co-researchers. Reviews have shown that participating in YPAR can improve skills related to agency and leadership, research, social skills, critical and social consciousness, and increase knowledge about the research topic [6,7]. Moreover, youth can improve their feelings of empowerment [6,8]. In a general sense, empowerment refers to: “[…] the ability of people to gain understanding and control over personal, social, economic, and political forces in order to take action to improve their life situations” [3] (p. 152). As such, YPAR can also be beneficial for the community and local organizations [7]. Actions initiated by youth are more aligned with the needs and wishes of their community [9]. YPAR has mostly been applied in school-settings to address problems with education, social inequalities, health, or the physical and social environment [10,11]. 

To date, top-down developed health promotion interventions have generally shown disappointing effects. This is especially worrisome for youth from low socioeconomic positions (SEPs), because health inequalities have been maintained or increased [12]. YPAR may be the key for developing more effective actions that match the characteristics and health needs of this hard-to-reach group [6,7,13]. ‘Kids in Action’ is such a YPAR study, in which academic researchers collaborate with 9–12-year-olds (further named ‘children’) from a low SEP neighborhood in order to improve their physical activity and dietary behavior. ‘Kids in Action’ was initiated because of the high prevalence of health problems in this particular community. We hypothesized that YPAR would lead to better tailored and thereby more attractive and effective actions for this low SEP community [6,9]. Throughout the study we tried to collaborate with children on the level of shared decision making—i.e., level 6 of Hart’s children’s participation ladder [14]. The overall research aim of improving children’s lifestyle was decided upon by researchers, but during the rest of the study, children were actively involved as partners: By actively giving input through participatory meetings, conducting research, analyzing results, and implementing and evaluating actions [9]. Children became partners starting with the participatory needs assessment, where children, parents and community professionals identified two main needs that specified the aim of the current study (i.e., improve physical activity and dietary behaviors). To gain a deeper understanding of the more and less effective elements of actions, as well as facilitators and barriers for sustainable implementation [15,16], academic researchers conducted an extensive process evaluation. Children were not involved as partners in the process evaluation, because it was conducted over several years and we worked with different groups of children each year. Moreover, the extensive process evaluation would take too much time, which children needed to develop actions.

Remarkably, few process evaluations of YPAR studies are published [17]. As participation of the target group in action development has recently gained increasing popularity, process evaluations of YPAR studies are urgently needed to gain insight into essential preconditions and challenges [18]. This article describes the qualitative process evaluation of ‘Kids in Action’ including the participatory process, the developed actions, and the outcomes of the study from the perspective of children and community partners. For a detailed description of the methodological process of combining YPAR with intervention mapping (IM) and how this was experienced by the researchers, we refer to Anselma et al. 2019b [19].

## 2. Materials and Methods

### 2.1. Study Outline

Researchers from the Amsterdam UMC collaborated with the sports-based daycare ‘Kids Aktief’ (KA) from the commencement of this study and the application for funding. Together with the local government, it was decided to focus on one specific community in their district that would benefit most from a YPAR study aimed at stimulating a healthy lifestyle in children. When setting up the study, the policy makers invited the research team to join an existing community project group working on healthy lifestyles of children in the community. Box 1 describes the most important participants and community partners of this study. The ‘Kids in Action’ study consisted of two phases, of which phase 1 started in 2015 with a participatory needs assessment. Research was conducted in collaboration with children, and interviews were held with parents and professionals from the community project group. Results showed that low levels of physical activity and unhealthy dietary behavior were perceived as the main health problems that children in the community faced. Therefore, improving these behaviors in 9–12-year-old children became the focus of phase 2, which started in 2016 and lasted for 3 years [20,21]. The current paper evaluates the process of phase 2, where children were involved throughout the process of action development, implementation and evaluation, from doing background research to developing and evaluating actions. YPAR was combined with intervention mapping (IM), a stepwise approach to developed evidence-based actions, to structure the process of action development and relate children’s ideas to evidence-based behavior change strategies [19,22]. Children were involved throughout the IM-process as much as possible, with certain IM-steps being adapted to be suitable for children. Some theoretical tasks were performed by an IM expert panel as they required specific knowledge that would be difficult or too time-consuming to teach the children. The process of combining IM and YPAR was iterative and is described in detail in Anselma et al. 2019b [19].

Textbox ACollaborations in the ‘Kids in Action’ study.**KA**: Kids Aktief—Community organization part of the Planning group.**Research team:** M.A., M.C., T.A.**YPAR group:** Child-researchers plus one or two academic researchers from the Amsterdam UMC (one being M.A.). YPAR groups were called an Action Team in year one and two, and the Youth Council in year three.**Action Team**: In the first two years of the project, there was one YPAR group at each of four primary schools in the community, called an Action Team.**Youth Council**: In the third and last year of the project there was one YPAR group in the community with representatives of the four primary schools. The Youth Council was hosted by M.A. together with three community organizations, and facilitated by M.A. and a social worker from one of these organizations.**Planning group**: Academic researchers from the Amsterdam UMC, youth policy managers of the local government of the North of Amsterdam, KA, child-researchers from YPAR groups.**Community project group**: Academic researchers from Amsterdam UMC, policy makers from the local government in the North of Amsterdam with a focus on youth, KA, representatives of other organizations working with children in the neighborhood such as social workers, teachers and principals of primary schools, and staff of after-school daycare organizations.**Respondents of interviews:** Three school principals, two school teachers, two policy makers from the local government with a focus on youth, four social workers from community organizations working with youth.

In the first year of phase 2 of the ‘Kids in Action’ study, we started a participatory group (YPAR group), called an Action Team, at each of the four primary schools in the community [21]. The Action Teams consisted of 6–8 children between 9–12 years old, who met weekly or every two weeks for 45–60 min. Children were recruited through the schools, and parents of participating children provided informed consent. Selection procedures varied per school and per year. For example, in the first year, at one school, all children could sign up; at two schools the meetings were held during school hours, therefore teachers selected children who could miss academic time; at another school, children who were part of the student council were invited to participate in the Action Team. Table 1 provides an overview of the members of the YPAR groups per year. Detailed information about the content of the meetings, the YPAR process and the developed actions is reported elsewhere [19]. In short, during the YPAR meetings, the Action Teams were trained in research skills and developed, implemented, and evaluated actions. In the first year, each Action Team conducted their own research to validate the findings of the participatory needs assessment. Program goals and performance objectives were developed and deliberated with the Action Teams. The Action Teams thought of ways to reach the goals, voted for the best ideas, and then further specified the ideas by making production and implementation plans. At the end of the first year, the developed actions were pilot-tested. At the beginning of the second year, new Action Teams were formed, with some children of the first year continuing their participation. One school decided not to participate in the second year because they chose to participate in another study. They decided to re-join the project in the third year. The three Action Teams in year two further implemented and evaluated the pilots of year one, but also worked on new actions. Table 2 provides an overview of the goals the Action Teams wanted to reach, the most popular initial ideas, and the developed actions.

Instead of one Action Team per school, in the third year one Action Team was started with representatives of three schools, called the Youth Council. The principal researcher (Manou Anselma) hosted and facilitated the Youth Council together with representatives of three community organizations and trained them in facilitating child participation. One social worker of one of the community organizations co-facilitated the Youth Council with the principal researcher throughout the year, with the goal of hosting the Youth Council by herself with a new assistant the following years. This collaboration between Manou Anselma and the three community organizations ensured the continuation of child participation in the policy of community partners after the research project. Children from the three highest grades of the four schools in the community could sign up for the Youth Council, but from one school no children signed up. The aim and the process was similar to that of the Action Teams of the first two years, but the Youth Council focused more on community actions instead of school-based actions, and had a broader focus than health.

### 2.2. Data Collection

Four main research questions guided the process evaluation: (1) How did children and community partners experience the participatory process and how will it be taken forward? (2) How did children and community partners experience the developed actions? (3) How did the involvement of children in decision making and community change, influence children’s health behavior and empowerment? (4) What are essential preconditions and challenges of YPAR?

We conducted a qualitative process evaluation because we were mainly interested in experiences and processes instead of numbers. Data was collected through focus group interviews with children and individual interviews with community partners. M.A. led the interviews and focus groups while a research assistant took notes, kept time and made sure all topics of the interview guides were covered.

#### 2.2.1. Focus Groups

The focus group interviews were held at the beginning and end of year two with the three Action Teams and at the beginning and end of year three with the Youth Council. No actions were developed in year one because the first year was mainly dedicated to completing the needs assessment. Therefore, the first focus groups took place at the beginning of year two. The focus groups had a semi-structured outline, where assignments were prepared beforehand and the facilitators asked guiding and clarifying questions. We used literature on empowerment theory to develop the protocol for the focus groups with children [3,23]. Through two group assignments where children had to think about actions in the community (details described below), the focus groups evaluated children’s individual, organizational, and community empowerment [3,21,23]. We used the definitions of Israel et al. [3], with ‘individual empowerment’ being defined as the children’s ability to make their own decisions and have control over their own life. We applied ‘organizational empowerment’ at the school level, and looked at whether schools enabled children to actively participate in decision making and whether schools were involved in community decision making. ‘Community empowerment’ was defined as the children’s and organizations’ ability to collectively reach the goals set for the community. The first assignment mainly assessed individual empowerment and awareness of the community. The Action Teams had to think of an action they wanted to implement and write down all the steps needed before implementing the action. For example, they were prompted to think about the goal of the activity, finances, getting approval, target group, promotion materials, etc. In the second assignment the children had to write down all activities and changes that had taken place in the community and at school in the last year. Of those activities and changes they identified if children were involved in the developmental process, how they felt about children being or not being involved, and how children’s opinions were valued. This second assignment assessed organizational and community empowerment. Because children’s participation and influence at school and in the community were assessed, the developed actions were also discussed. In the last focus group of the Action Teams, children could also bring friends to add information about empowerment of non-Action Team members.

#### 2.2.2. Interviews

Towards the end of the study (May–June 2019), we reflected on ‘Kids in Action’ through individual interviews with the community partners who most closely worked together with the research team. The potential respondents were contacted via e-mail or phone. All community partners agreed to an interview, except for one teacher and one school principal who did not have time.

The interviews with community partners were guided by the RE-AIM framework [24]. The RE-AIM framework aids structural evaluation of actions on five dimensions: reach, effectiveness, adoption, implementation, and maintenance [24]. ‘Kids in Action’ can be seen as a participatory process leading to a multicomponent action. In order to tailor RE-AIM to the participatory process of our project and to be able to answer our research questions, we adapted the interpretation of the five dimensions and added new dimensions such as empowerment, collaboration, and communication. Table 3 presents examples of questions from the interview guide. The interview guide consisted of seven sections: (1) Reach—which children participated in the actions and the YPAR groups; (2) Effectiveness—both of the perceived effectiveness of the actions on health behavior and the effectiveness of the YPAR method on, for example, empowerment; (3) Adoption—of the actions and the YPAR method in the schools and community project group; (4) Implementation—of the actions and the ‘Kids in Action’ study in the community; (5) Maintenance—of the actions, the perceived effects and the YPAR method; (6) Collaboration—with the principal researcher, KA, and within the community project group; (7) Communication—between the principal researcher, children, parents and community project group, and communication in general.

#### 2.2.3. Analyses

The focus groups and all but one interview were audio recorded and transcribed by the academic researchers who conducted the interviews and focus groups in Dutch. One respondent did not feel comfortable with the audio recording, and therefore the assistant took extensive notes. The interviews were summarized and sent back to the respondents for a member check. M.A. coded and analyzed the transcripts of the focus groups and interviews, and the extensive summary of one interview, in ATLAS.ti. Empowerment theory was also used to develop a starting list of codes for empowerment [3,23] (Table 4), which was made by M.A. and checked by T.A. Provisional coding was used for everything related to empowerment, both in the focus groups and interviews. Extra codes could emerge and relations were derived from the data. We used evaluation coding as a basis to code topics other than empowerment, and combined this with several other coding methods such as magnitude coding, descriptive coding and emotions coding, to be able to differentiate between positive and negative items, to label emotions and experiences, and to be able to give value to the variety of materials [25]. M.A. conducted all coding and, where needed, consulted TA to resolve uncertainties. All analyses were conducted in Dutch and results were translated by M.A. for the current paper. Quotes that could serve as an example of, or explain a piece of text, were also selected and subsequently translated by M.A.

## 3. Results

Table 5 describes the characteristics of the focus groups. Most focus groups were divided over two sessions as the content did not fit within one session. Table 6 provides an overview of the respondents of the interviews and their occupation. The results are presented in the following themes: empowerment, becoming part of the project, collaboration, reach, effectiveness, adoption, implementation, and maintenance. The section ‘empowerment’ mainly consists of data from the focus groups with children, the other themes mainly consist of data from the individual interviews with community partners.

### 3.1. Empowerment

Table 7 provides an overview of the coding scheme used for items related to RE-AIM and the YPAR process.

#### 3.1.1. Individual Empowerment

##### Critical Awareness and Critical Thinking

AT1 and AT2 already had a good sense of community at the beginning of the year, took preferences of other children into account, and could think about the bigger picture [AT1, AT2]. For their idea to create signs that people had to clean up after their dogs, AT1 thought about financing, involving the community to increase awareness, promoting their cause in the community and impact: “*But when you make a sign, people just walk by and do nothing. It has to draw attention. It should not be that you have woods or a park and you put up a sign and everybody just walks past it and doesn’t look at it*.” [AT1]

AT3 and the Youth Council started as new groups in the beginning of the school year and they had to get used to the participatory approach. It was more difficult for them to think about the bigger picture. AT3 could only think about their own individual network, for example by only inviting their friends to participate in actions and asking help of their own parents. They took the opinion and preferences of other children into account, but could not think of practical steps how to involve them. Also, the Youth Council was not aware of any community organizations they could ask for help. At the end of the year, their thought processes were much more clear and practical. The Youth Council was for example more aware of the costs of activities, e.g., using second hand materials, and AT3 had more fruitful discussions regarding how to develop activities and create realistic ideas.

Children from all participatory groups indicated that they preferred to ask approval for an action quite early on in the process, to prevent investing much effort and time and then not getting permission to continue with it.

Two principals described that in the beginning children started super enthusiastic and when they got an assignment they went all out and did not think about feasibility or practicalities [S1, S2]. During the year, they learned to take into account the bigger picture, for example, to include more people, cultures and viewpoints [C8].

##### Learning

Most children mentioned that they learned more about healthy behavior, for example about physical activity guidelines and the importance of a healthy diet [AT1–AT3, YC]. They also learned to discuss within the group, collaborate with others and do research. Children really liked doing research, such as interviewing their peers and making pictures.

One of the teachers observed that children of the Action Team became leaders in class as they felt important and more confident [S5]. Respondents also mentioned that children learned skills that could benefit them at school and outside of school, became more aware of themselves and the community, improved their collaborating skills and making democratic decisions, gained more confidence, and became representatives for other children in the community [S1, S5, C9]. “*Before he* [member of Action Team] *never stood out, but he left school much more aware of his skills. That he can ask things, be critical to himself and to others. Also some of the other kids. I can remember two or three girls, they became more capable to find common objectives through collaboration. I’ve seen that in at least three or four kids.”* [S1] S3 further mentioned that by participating in such a group where children had more space and attention, they developed and showed talents that may be missed in the classroom. Respondents especially enjoyed seeing the improvements in children who they at first did not expect to participate in such a group [S1, S3, C8]. C8 and C9 heard back from parents that their children were happier at home, were more talkative and had gained more confidence, through participating in the Youth Council [C8].

##### Participation

Children of the Action Teams felt special because they were members of the Action Team, as indicated by one of the Action Team members being annoyed with one of the non-Action Team members, because she “*did not understand the process*”. Both Action Team and non-Action Team members felt they had participated in organizing activities [AT1, AT2]. Children liked actively working on developing and implementing activities most [AT1–AT3, YC]: “*I liked everything but mainly implementing everything and sitting around the table and develop things and that you saw that children really liked it, such as the Olympics and making the dog-poo signs and asking questions in classrooms and the cooking workshops*” [AT2]. Children liked being appreciated for the actions they had developed and believed they did important work for the community [AT3, YC]. They liked the activities they had developed, as activities organized by adults were ‘old-fashioned’ [AT2, YC]. The Youth Council felt that they could really add value as they had more and better ideas than adults: “*Because children are more creative and think and look at things in a different way than adults, and because of that can make things more fun. And children can think in a smarter way than some adults when they are being difficult. They* [adults] *only think about one thing: what the problem is. But children can go around that and find a better solution*.” Children mentioned that they felt strong, proud or ‘the boss’ when they helped to organize something [AT2, AT3], such as the Olympic sports tournament: “B*ecause when we arrived I thought: God, I decided on this*!” [AT3]. However, not all Action Teams felt ownership over the developed actions, such as AT2, who could not come up with all the actions they had helped to organize.

One of the keys to the success of the project was that children liked participating in the participatory groups because they were given much freedom, they felt listened to and felt valued for their ideas, as well as seeing results from their involvement [S2, C8]. Because the activities were developed by children, they were also more supported by other children [S3].

##### Self-Efficacy

AT1 and AT2 were confident about doing everything themselves. If required, they sought help from the facilitator and other teachers of KA. AT1 did not want too much help, because: “*It’s also not cool if somebody else does it while they are our ideas and we can also execute them ourselves*.” At the end of the year were even more confident in talking to community organizations, potential sponsors, teachers or the principal: “*Miss can I say something, I don’t think we need any help. And do you know why? Because we are* [the Action Team]*. We did a lot of things by ourselves.*” They could easily think of ways of persuading others to help them, for example by telling the school principal that the ideas of the Action Team match the objectives of the school. They also discussed what they would do if the principal would not approve of their idea: they would just come up with another idea.

AT3 needed much assistance at the beginning of the year in thinking about steps to take and how to execute them. At the end of the year they had more ideas and more confidence in executing them. In the Youth Council, most children felt they could do most things themselves, which was not always realistic. At the end of the year, they had become more realistic and for example said they could ask the local government for approval, but would like the facilitator to go with them.

In all participatory groups children immediately started thinking about ideas for actions and implementation. Through the YPAR process they learned that there were multiple steps they had to take and thought of more feasible ideas and timelines [AT3, YC]. Children also noticed improvements in their self-efficacy. At the end of the year a boy of AT3 said he had learned to speak up more and a girl of AT2 who was shy at first, confidently said that she had administered questionnaires in several classrooms by herself. Teachers mentioned that children from the Action Teams had really developed during the year, dared to ask more questions of teachers and the principal, and have discussions with them [S1, S4, S5].

#### 3.1.2. Organizational Empowerment

##### Role of the School

Most children had positive relationships with their teachers, which was important to get their research done or actions implemented [AT1, AT2]. Mostly it was principals who played an important role in the project, because children had to ask permission to organize an activity at the school. The principals were always willing to talk with the children and explain why they supported an idea or not [S1, S2, S4]. Other day-to-day tasks were mostly handed over to teachers [S2, S4].

It was felt that schools could be more involved in the community [G7, C9, C10], “*because it* [activities in the community] *fails or succeeds based on the commitment of the schools*” [S1]. S1 explained that if there was a community meeting where schools were invited, they were the only school that was present, which was considered to be a shame. The schools and the principals have an important role in the community, which was not considered as such by all schools [S1, G6, G7, C9–C11].

##### School’s Value of Children’s Opinion

The Action Teams mentioned that child participation in school decision making was low. AT3 felt that their opinion was often asked by the school, but they were upset because nothing was done with their opinion: “*They ask our opinion, but then they don’t do what we asked. Then I just become angry*.”. They felt that they were rarely taken seriously because they were children [AT1, AT3]. S1 acknowledged that adults often wrongly assume that children are not able to do certain things. At the end of the year, the Action Teams indicated that they had participated more in decision making at school, but that was mainly through the actions they had developed. AT3 described that when they wanted to fix something at the school playground, the principal organized a sponsored run for them to get the finances for making the playground improvements. At this school, two kids of the Youth Council also started an anti-bullying club that was very successful and had many members [YC].

When an Action Team or the Youth Council had tasks that needed to be executed, the schools gave the children much responsibility with teachers/principals just guiding the process [S1, S3–S5, AT1]. However, the community partners said that the Youth Council came up with many ideas for actions for their schools, but that many were not supported and implemented by the school [C6–C11]. They would like schools to be more involved in the community and support the children’s work: “*Children have fantastic ideas, but our organizations can’t implement them all. Schools are not involved, so at the end of the day children take their ideas home and nothing is being done with it*” [C10].

#### 3.1.3. Community Empowerment

##### Involvement in Decision Making

In the beginning of the year, children felt that they had more influence in decision making in the community than at school [AT1–AT3, YC]. AT3 had the least positive experiences in decision making with community organizations; at this school, fewer activities were organized by community organizations. At the Youth Council the children felt they had some say in activities of community organizations, but not much. When they were asked about activities where children did not participate in the development, it was “*too much to write down*”. One of the children explained that children should have more influence in decision making: “*It is for the better, because we are also here and it is important that we can also make our voice heard. Not only adults can make good decisions, we as children can do that as well. That’s why we try to make our voice heard* [through the Youth Council].”

At the end of the year, all groups felt that children were more involved in community decision making, especially through the activities they had organized themselves. Also non-Action Team members felt they had a voice in decision making and were satisfied with the participation of children in the community [AT2]. The Youth Council became more aware of all the community organizations and also felt that the Youth Council was taken more seriously. During the year the local government often invited the Youth Council to meetings to include children’s point of view, which community partners appreciated. Despite this, some community partners were skeptical about how much of the children’s ideas were going to be implemented by the local government [C9–C11].

### 3.2. Becoming Part of the Project

Respondents’ reasons for participating in the project were because the project had similar goals to their school/organization or because it was decided for them on a higher level [S1, S2, G7, C9]. In the community all principals decided to join as it suited their schools’ policy and plans [S1–S4]: “*Joining the project was something we consciously chose for as we saw it as added value*.” [S3]. The policy makers further explained that the project related to their policies aiming to improve children’s health behavior and as it was a practical research that could strengthen the community [G6, G7]. The policy makers had successfully collaborated with KA before and because they were approached by them they were more inclined to agree.

Most respondents did not have any expectations when joining the project [S1–S5, C9, C10]. They explained that the community had seen many projects passing by without any useable outcomes so they stopped having expectations: “*We don’t have any expectations anymore and then any benefits that come from a project are perceived as a win*” [S5]. S5 mainly hoped that children liked participating and with their enthusiasm could reach other children. C11 mentioned that she had high expectations of the Youth Council because it was hosted by so many community partners. The local government expected positive changes in the community because they knew KA as a hands-on organization and they were confident that research backed-up by them would be a good investment for them and the community [G6, G7]. Also, the grant application was judged to be comprehensive and substantial [G6, G7].

### 3.3. Collaboration

All respondents were positive about the collaboration with the research team. However, some community partners and the research team had to get used to each other and their different ways of working [S2, G6]. “*But what I mainly noticed is that if we as a school thought that something was not practical there was always room to talk about it and figure out: How can we do it in a better way?*” [S2]. Some members of the community project group were a bit hesitant about collaborating with KA, mainly because in the beginning they were not aware of the goal of the project [G6–C9]. After that became clear and they realized the project only wanted to strengthen already existing work processes, strong collaborations were formed [G6–C9]. The community partners that co-hosted the Youth Council worked closely together with M.A. and appreciated the collaboration [G7–C9]: “*I’ve never worked together with someone who… who I can trust, you know. I know that if I could not handle something, I was listened to and that was a huge support. No I really think you’re great. I am going to miss you. Really, really. You’re a person close to my heart*” [C8]. C9 specifically appreciated that they were invited to participate in an action through which they received sports materials so the children at their organization could be more physically active. For them this was a perfect example of how different organizations and projects could strengthen each other [G6–C11]. This was sometimes a challenge in the community, as there were many organizations with their own agenda [C9–C11].

Schools appreciated the regular contact with M.A. which helped them to stay updated and they appreciated the reminders if they forgot to respond [S1–S3]. Respondents liked the quick response via e-mail or telephone and that M.A. communicated ahead of time regarding the planning [S3–S5, C8–C10]. Every two months there was a meeting with the research team and the local government, where—without an agenda—the status and challenges of the project were discussed [G6]. This was valuable for the local government to stay updated and help out where needed, and was experienced as crucial for the strong collaboration between the local government and the project [G6, G7]. They explained that the research team could have asked more from them, for example, that they could have already from the beginning been involved in maintenance of the actions.

### 3.4. Reach

M.A. led the research for four years, which meant that all children really got know her. This helped to get children involved and keep them involved in the project [S5–C11]: “*It is a tool, having a familiar face in the community. If something is up with them, then they know who to go to and they know you are watching them and looking after them. Then I think, yeah, you know, we should use this more in such communities. Especially when I now look in this community, I think about the familiar faces working in the community*” [G7]. Recruitment for participants was easier when the school principal encouraged teachers to recruit children [S1, S5], and when a researcher went from class to class to explain the research and hand out attractive leaflets for the actions [S3, G7, C10]. In the Youth Council children from various ethnic backgrounds participated and more boys than girls, which was a surprise [G6–C10]. It was considered a shame that not more children could participate, as more children could benefit from participating in such a project [C10]. Having one Action Team per school—versus one Youth Council—had the benefit that more children could participate [G7]: “*But I don’t know how realistic it is to maintain that. If you indeed have one Youth Council in the* [name community]*, that would be amazing*.”

Respondents did not know whether the actions indeed reached children with unhealthy behaviors [S1, S2]. All respondents were satisfied with the number of children that was reached, especially with the extracurricular sports activities, with about 60 children participating weekly, and in the Olympic sports tournament, where 350 children participated.

### 3.5. Perceived Effects

The main perceived effect of the project was that children and parents became more aware of the importance of physical activity and healthy dietary behavior [S1–G6, C8, C9]. Real behavior change was more difficult to perceive and respondents were unsure if this was reached [S2–G7, C10]. Another important perceived effect was that children and community partners learned from the project (for children’s learning see Section 3.1) [S1, S5–C8]: “*Well I at least became a lot more aware. Look, you actually have known your whole life that you need to give children a voice in things and that you have to actively involve children. And the entire education system of course has also developed from more whole-class teaching to a lot more interactive and together. But yes this should happen in more areas. That is the good thing about it, that even if you are old and grey* [laughs] *you can still learn certain things and develop*” [S1]. C8 described that because they facilitated the Youth Council together with M.A., they learned much about how to work together with children in such a project. It was also mentioned that being involved in the project helped to place healthy behavior higher on the agenda and strengthen already existing policies [S2, S3, S5–G7, C9]. As the Action Teams and Youth Council were so successful, it served as an example to take child participation more seriously [G6–C8].

Different factors facilitated the positive effects. First, because M.A. spent a considerable amount of time in the community and was always present at meetings within the community, a strong network was created including children, parents and community partners [G6, G7, C10]. In the beginning, community partners and policy makers perceived a good relationship with schools as being challenging, but the good relationship with schools became a strength [G6]. Second, the project was able to reach many children from the start, and those children were so enthusiastic that they spread the message to other children and parents [S2, S5–C8, C10]. Third, the research team and the collaborating partners really listened to the children, worked together with them on an equal level and took action [G6, C8]. Furthermore, most implemented actions were a success and created more support for healthy behavior and child participation in the community [G6–C9, C11]. Lastly, the strong collaboration with community partners was mentioned as a strength of this project [G6–C11]. For example, because the project was embedded within KA, there were always enough people available to help in executing actions and measurements. Therefore, these activities were not a burden for the school staff [S4].

### 3.6. Collaboration with Community

For children, community organizations, teachers and parents were their most important collaboration partners [AT1–AT3]. AT1 and S3 mentioned that parents of the parent committee at school were known for being involved in the community. They for example helped with activities at school and helped children with organizing activities. C8 explained that children participating in the Youth Council became very involved in the community and were often organizing/participating in community events. They proudly represented the Youth Council, and with that, inspired other children: “*When children are involved in something as the Youth Council, they have more confidence and because of that they are happier and participate in more activities. They radiate this onto others. You know they draw more children to these activities, those kind of things. Those are all positive side-effects.*” [C10]

### 3.7. Adoption

Generally, respondents from the schools mentioned that they were well informed and therefore knew about the goals and processes of the project [S1–S3, S5]. Some respondents mentioned that providing more updates [S1, S3–S5, C10] and more communication about the project within the schools [S2–S5], could have supported the adoption of the project: For example, the contact person within the school could share more information within the team and engage teachers in activities [S1–S3]. “*As a teacher you sometimes have the feeling that you are less involved or don’t have an active role in a project. I didn’t really have the feeling that I had an active role. And now that I think about it, maybe I would have liked to have that. When the school is actively involved, it also becomes more of a school project*” [S5]. Since schools have a busy program and a shortage of teachers, an extra project does not automatically receive attention [S1–S3]: “*Sometimes we have to push teachers, because teachers do think it* [the project] *is important, but so many other things are also important*” [S4]. Another factor hindering adoption at one school was a significant amount of change in the management [S5]. The successful implementation of actions may have supported adoption [S2, S3, G6] and involvement of teachers [S2].

### 3.8. Implementation

#### 3.8.1. Process of Collaborating with Children

The schools mainly spoke about the process of the Action Teams of the first two years, as the Action Teams met at the school and organized mainly school-based actions [S3–S5]. The Youth Council met in a community center and therefore teachers felt less involved. S5 explained that one version was not better than the other, but the Youth Council was less visible for the school: “*The first year I thought it was great; everything was clear. The kids came together and they told us what the meeting had been about. But that became a bit less in the last year. […] I have the feeling I did not see children participate as actively as in the first years and I didn’t see as many actions. But that could also be because the meetings were not in the school anymore.*” The value of the Youth Council was that children learned to cooperate with children from other schools [S5, G7]. As the Youth Council was hosted by multiple community partners, there was much support for developing and implementing actions from the start [C10]. The Youth Council became very popular in the community and was invited to many community meetings [G6, C8–C10]. As a consequence, children had less time to work on the development and implementation of their own ideas, while that is what kept them involved and enthusiastic [S2, G6–C11]. Children mentioned that they would like to have a less busy schedule and the facilitators saw that children became less involved towards the end of the year [C8–C10].

Some points for improvement were suggested in the process of collaborating with children. First, the Action Teams and Youth Council could involve more children in the decision making process, making it a project of all children instead of only the ones in the participatory groups. Second, it was considered a shame that one school did not participate in the Youth Council, due to extracurricular activities happening at that school at the same time as the Youth Council’s meetings [S4, C10]. For the following year the schedule of the Youth Council should be better suited to all schools. Third, the low parental involvement was mentioned as a point of improvement, but was acknowledged as a challenge and time investment [S3, S5, G7, C10, C11]: “*Maybe we could have invested more in parent participation. I think that should also be a focus point for the Youth Council for next year. With the cooking classes we managed to have some parents present, but it is very difficult and you have to be lucky to find a mother who is interested. But it is definitely something we should focus on*” [C11].

#### 3.8.2. Actions

C9 believed that the project definitely helped to create sustainable actions. Respondents mentioned that children were very enthusiastic about the developed activities. Two of the school principals explained that at the after-school sports activity at their school, more children joined every week and they were happy with this development [S2, S3]. That was also why it was so important that activities were continuous throughout the school year(s), so children and parents knew what to expect and where they could go [S1–S3].

All schools were positive about the Olympic sports event that was organized for children from all schools in the community and they hoped that it would continue to be a yearly event. They mentioned that it brought schools and children together, which was good for the community. S3 was proud of the water policy that was successfully implemented at her school, despite children not being happy with this policy in the beginning. Also one of the boys who helped to implement the policy did not always adhere to it. But when the teachers told him “*Hey, this was your idea*!”, he acknowledged that and without discussion threw his juice away. Most schools did not know much about the cooking workshops, but heard positive stories from the children [S2–S5]. C8 was only involved in the project in the last year, but had heard about the project’s actions through parents, who were always very positive. Parents said they liked that the activities were within the community because that was exactly what was needed. In the beginning of the project G6 and G7 felt that it took a while before actions were implemented. When the research team explained the process of participatory research again it made them understand that this was a different process from how the local government usually works [G6, G7]. G6 and G7 suggested organizing some small activities in the beginning of the project: “*Of course you should not do large activities, because yes first you want to know what they want, but you know, some small activities that children can influence and that you are going to organize. Then you involve the schools* [and they feel] *like, ok well ok, it is going well, everything is going well, it is a trustworthy organization. Then you’re accepted faster and other organizations have less cold feet to collaborate*.” [G6]

Children liked the implemented activities. In the Action Teams they mainly discussed the cooking workshops and Olympics, as most children had helped in the development. They liked the practical cooking workshops and going to the local garden [AT1, AT3]. They hoped that future cooking workshops would include more foods they liked and that everything was not super healthy. They also wanted to introduce food from their culture [AT1]. Most kids were also positive about the Olympics but it was also mentioned that there was a significant amount of cheating and rough play between schools, which they did not like [AT1, AT2]. Children were very proud of the signs that they made to clean up after your dog and that it was actually implemented by the local government. The Youth Council was positive and proud of the Youth Club center they created and hoped it would last for many years.

### 3.9. Maintenance

Maintenance was an important topic for the respondents. They hoped that the line of work—both with the active participation of children and the actions—would continue. But all had doubts about how this would be taken forward. Mainly finances were a perceived barrier. S2 and S3 wanted to continue the extra sports activities at their schools, but questioned who was going to pay for it. Respondents mentioned that it could have helped sustainability of the actions if the research team would have pressed thinking about sustainability at an earlier stage [G6, G7, C11]. Some actions were taken up by community partners: the water policy that was introduced was sustainably implemented [S3, S4], some sports activities received extra funding [G6, G7], and the cooking classes and Youth Club center were taken up by community partners [G6–C11].

The community network was perceived as a facilitator to continue working on child participation and healthy behavior [S1, C8–C11]. At different organizations in the community they valued child participation [G6–C8, AT1, AT2], and also at schools they were trying to incorporate participation in their curriculum [S1–S4]. The plan was to continue the Youth Council [G6, G7, C8–C11]. Respondents enjoyed seeing the children so passionate and involved in their community[G6-C8]. Acknowledging the investment in time, energy and budget, they considered it a challenge to find someone to take over the facilitation role of M.A. [C9–C11]: “*It actually shocked me that you spent a full day a week on it* [the Youth Council]*. So I thought, yes, you know that’s actually quite a lot*.” [C11]. The community partners co-hosting the Youth Council were also dependent on funding from the local government [G6–C11]. During the course of this study, participation of children was always on the agenda of the local government, who were therefore very supportive of the study and children’s activities [G6, G7]. But with someone else in charge of policy in the future, this may change [G6, G7, C9–C11].

At the end of the interviews we asked the participants to pick three attributes that they valued most about ‘Kids in Action’. We presented eight items, but respondents could also add items themselves (Table 8). ‘Participation of children in decision making’, ‘the developed actions’ and ‘communication’ were mentioned most often.

## 4. Discussion

This study evaluated the process of the ‘Kids in Action’ study. An identified strength of ‘Kids in Action’ was the strong collaboration of researchers with both children and community partners. Through YPAR, children became empowered to take action in their own community, as also noted in previous studies [26,27,28]. We experienced that when working with this age group, a strong collaboration between children and adults is recommended. Children cannot execute all their ideas by themselves, but need the help of adults [29], i.e., the participatory group facilitators and relevant community partners. Community partners enjoyed discussing children’s ideas and working with them towards implementation. Because of the positive experience with children’s participation, community partners wanted to continue with this after the study. A recent review found that through being involved in YPAR, adults appreciated children’s abilities and valued their participation [11]. This is one of the most valued outcomes of this study, as we feel that children should not only participate in decision making during research. A precondition for—sustainable—participation in the community is training community partners to actively engage children in decision making, for example using readily available manuals on child participation [6]. In our study we trained the community partners who co-hosted the Youth Council, so they could continue facilitating children’s participation in the community.

Children appreciated and enjoyed being part of action development, as it created the feeling that it was ‘their’ action. This feeling was shared by community partners who for example mentioned that the water policy was well supported at school because it was the children’s idea. Children felt the actions better-suited their interests because children had co-developed them. For example, the ‘dog-poo’ signs were appealing to children because they were drawn by children and colorful. A recent review also found that engaging youth in inquiry aided the research process, for example because youths can stimulate each other to participate in actions [11]. By developing and implementing actions together with children, ’Kids in Action’ tried to work on the level of participation of shared control and decision-making power between children and adults [14,27,28,30]. On this level children are taken seriously, they work on actions and see their own actions being implemented [18]. When children are involved in advocacy and organizing, this can lead to environmental outcomes such as changes in peer norms and program development [11]. It should be kept in mind that sharing power may not be possible for every decision [31]. To preserve scientific integrity and methodological quality, researchers need to guarantee adherence to guidelines regarding scientific integrity and methodology and in the case of our study align the developed actions with health behavior theories [19]. On the other hand, when children feel strongly about something that researchers do not recognize, children could be allowed to go through with it. Through YPAR children and researchers educate each other [28,29]. For example, with the cooking workshops, children wanted a combination of healthy and unhealthy foods, where the involved adults wanted to focus on healthy foods only. Through dialogue, children came to understand that the focus had to be on healthy foods as that was the goal of the project and children helped adults to design attractive leaflets and content of the workshops that would appeal to children, i.e., healthy meals that children enjoy. However, when children evaluated the actions, they seemed to had forgotten this process and felt they had been limited by the boundaries of healthy foods. This shows it is important to constantly reflect with the children on the goals of the study and together create a clear understanding of the responsibilities and rationale [18,28].

Community partners perceived that ‘Kids in Action’ contributed to more awareness of children about healthy behavior though they doubted whether the actions had led to actual behavior change. School staff indicated that more involvement of parents might be necessary to reach behavior change, as children of this age are largely dependent on their parents in their health choices. In the present study, parents were involved in the needs assessment but they did not want to have an active role in action development. Future research should examine how parents can be motivated to actively participate in community-based research. According to all community partners the largest effect of ‘Kids in Action’ was realized on children’s empowerment. Reviews looking into outcomes of YPAR found similar results, with positive effects on agency and leadership, social, interpersonal, and cognitive skills [6,7]. It was interesting to see in the current study that children who had already participated in a YPAR group during the previous school year were more experienced in the YPAR process than children who just joined. They had already developed more critical thinking and awareness of the community and were more realistic in their ideas. Thus, including more children in the participatory process and for a longer time period—for example, by incorporating it in the school curriculum [30,32]—may lead to empowerment on a larger scale. Furthermore, to optimally develop children’s empowerment, the focus should not only be on the individual. According to empowerment theory, individual, organizational and community empowerment cannot be interpreted separately [3,23]. Therefore a systems approach, in which the system around the child is influenced would be optimal [23,28,33]. In the current study, children’s empowerment developed on an individual and community level, but less on the organizational level. A possible explanation for this is that by being part of the Action Team or Youth Council, children individually developed. Furthermore, community empowerment developed because they often collaborated with community partners who valued their opinion, also in decision making that was not related to ‘Kids in Action’. However, as schools stated from the beginning of the project that they were very busy and endured a shortage of teachers, we did not ask too much involvement from the schools throughout the project.

Community partners mentioned many preconditions for a successful YPAR project, of which a close collaboration of researchers with children, KA, local government and community project group, was indicated as most important. Collaborations between academics and community organizations may provide an enabling environment for successful YPAR [34]. Crucial in ’Kids in Action’ was that the community partners were interested in child participation and understood the power dynamics between children and adults in YPAR, which helps gain support for such a study [18,35,36]. Community partners were willing to collaborate with children to develop their ideas for actions and when necessary integrate it in their organization’s policies. As mentioned before, schools were less involved during the study. A learning point was to have more open communication with schools about their role and expectations. Schools mentioned that they would have liked more updates and involvement during the project. In retrospect, we should have openly discussed their level of involvement instead of assuming that they were too busy to be involved. A precondition and also a major challenge for closely collaborating with children was to organize the meetings in such a way that children stayed interested and motivated, and maintaining a trustful relationship [36]. Tools such as icebreakers and creative assignments facilitated this, as well as having conversations unrelated to the research if children had something they needed to share [37]. We should also keep in mind to not ask too much from children, which some children experienced in the Youth Council. Future studies could create the research agenda and planning of the YPAR meetings together with children. Another precondition for YPAR—as experienced in all YPAR—is a considerable time investment of the principal researcher, the children and community partners, to be able to create close ties, hold regular participatory meetings, and co-develop and implement actions [35]. It also required high flexibility from researchers and the community project group, as YPAR remains an iterative approach dependent on many uncontrollable factors and school schedules [34,35].

This study includes several strengths and limitations. Even though we included the views of different community partners, we only evaluated the study with those who were closely involved, which could have biased the results. Also, we did not include children in the initial design during the grant application phase. This could have benefited mutual understanding between researchers and children from the beginning, stating boundaries and discussing where concessions can be made [7,38]. However, applying for funding is a lengthy and uncertain process. Recruiting schools for a project planned in future school years that might not even get funded is challenging. Furthermore, we did not conduct the process evaluation with children as partners, because it did not fit the timeline of the study and would be an additional task and time investment for the children. Future YPAR studies may explore ways to evaluate the entire research process together with children as partners. This may provide additional relevant insights to strengthen YPAR studies. A limitation of the ‘Kids in Action’ study is the low involvement of parents and as a result we also did not include them in this process evaluation. For future studies with more resources and personnel, it would be valuable to also include parents. Furthermore, not all focus group interviews had the same depth. In some groups not all children participated fully, for example because they were tired or distracted, which negatively influenced their input in the assignments. A last limitation is that data was not independently coded by two researchers due to time constraints. The coding schemes were however checked by a second researcher, who was also consulted in case of uncertainties. A strength of this study is that through our different types of data and coding strategies, we managed to give value to much of the participant’s input, opinions and feelings. Second, this study provides much information about how to collaborate with children and community partners in YPAR, as all who closely participated in ‘Kids in Action’ were included in this study and shared their experiences. Third, the interview and focus group guides were based on the RE-AIM framework and empowerment theory, which gave structure to this process evaluation. Another strength is that focus groups were organized at the beginning and end of two school years, allowing to evaluate changes in children’s feelings of empowerment and experiences with YPAR. Moreover, in the first year, both children who did, and some children who did not participate in a YPAR group joined the focus groups. As empowerment was evaluated on different levels, an added strength is that it gives more insight into how the influence of YPAR on children’s empowerment was constructed.

## 5. Conclusions

In the ‘Kids in Action’ project, actions to promote physical activity and healthy dietary behavior were developed, implemented and evaluated together with children from a low socioeconomic community. The project was well adopted in the community, which can be ascribed to the strong collaboration between researchers and community partners and the willingness of community partners to participate with children in action development. This led to co-created actions that suited children’s needs and interests and where possible were embedded in schools or other community partners. Maintenance of child participation and the developed actions after the study was strived for by community partners, but considered a challenge because of finances and politics. Children liked seeing the results of their weekly participation, and being involved in YPAR improved their empowerment. A recommendation for future research is to integrate child participation in policies of community organizations and schools, in order to reach more children. Closely collaborating with schools and parents can further benefit support for the co-created actions and increase the effects on health behaviors.

## Figures and Tables

**Table 1 ijerph-17-00625-t001:** Composition of Youth-led Participatory Action Research (YPAR) groups.

	Year 1: Action Teams, No Focus Groups	Year 2: Action Teams (AT)	Year 3: Youth Council (YC)
**School 1**	Seven children (6 girls, 1 boy), weekly meetings after-school hours of 45 min, followed by 45 min of sports. Children were already part of their school’s student board.	**AT1** Eight children (7 girls, 1 boy; 2 from previous year), weekly meetings after-school hours of 45 min, followed by 45 min of sports. One of the new children was chosen by a teacher, two were chosen by the 2 children from the previous year, the other children asked if they could join.	Five representatives (2 girls, 3 boys).
**School 2**	Six children (2 girls, 4 boys), every 2 weeks a meeting of 1 h during school hours. Teachers chose the children.	**AT2** Six children (2 girls, 4 boys; 4 from previous year), every 2 weeks a meeting of 1 h during school hours. The 2 new children were chosen by the teachers.	No representatives as no children signed up.
**School 3**	Six children (1 girl, 5 boys), weekly meetings after-school hours of 45 min, followed by 45 min of sports. All children from grades 6—8 could sign up.	**AT3** Six children (2 girls, 4 boys; all new), weekly meetings of 45 min, followed by 45 min of sports. All children could sign up.	Three representatives (1 girl, 2 boys).
**School 4**	Six children (3 girls, 3 boys), every 2 weeks a meeting of 1 h during school hours. PE teacher chose the children.	School did not participate.	Five representatives (1 girl, 4 boys).

**Table 2 ijerph-17-00625-t002:** Overview of goals, initial ideas, and the developed actions of the Action Teams (adapted from [19]).

Study Goals	Initial ideas that were voted for	Actions	Implemented by YPAR Group and:
Children always eat a healthy (amount of) breakfast in the morning before they go to school.	Create a series of lessons on what a healthy breakfast is and combine it with quiz elements, then children will like it more and remember the message (school 1, 2, 3).	Cooking workshops. At first for one month, then they were continuously implemented by another community partner.	KA and other community partners
		Quiz at school (once) and regularly recurring during after-school activities.	School
Children eat less unhealthy snacks at school.	Organize a competition at school where children can win a prize if they take healthy snacks and lunch to school (every month a different prize) (school 2).	Healthy snacks and lunch competition at school during three months.	School
		Cooking workshops. At first for one month, then they were continuously implemented by another community partner.	Community partners
Children drink only water at school.	Create a water fountain at the school playground where children can always drink water during and after-school (school 1, 3) Start a policy at school that children can only drink water (school 3)	Water fountain installed, together with a policy that children can only drink water at school (school 3).	Local government, school
Children drink tea without sugar.	Create a lesson series where children learn to drink tea without sugar, then they will get used to it and like it (school 1).	Cooking workshops. At first for one month, then they were continuously implemented by another community partner.	Community partners
More children play outside (actively).	Make playgrounds with equipment suitable for children of different ages and teach children active games that they can play there (school 1, 3).	The local government adjusted several already existing playgrounds.	Local government
		After-school activities. Organized for four months where children learned new active games that they could play.	KA
		Promotion of active games. During after-school activities, more focus was placed on active games that children themselves could play without many extra materials.	KA and other community partners
More girls participate in after-school sports activities.	Organize more girls-only activities, and ask girls what kind of activities they like (school 1, 2).	Weekly girls-only activity. Started for the duration of two years.	KA
Children participate in sports activities of their own preference.	Let children co-organize activities and make sure there are good coaches to supervise, so they will like it more (school 1, 2, 3).	Olympic sports event. Consisted of after-school sports activities followed by a sports tournament for the four schools in the community.	KA
		Children co-organized all intervention activities.	Research team
		Co-deciding on after-school sports activities. Activities with the aim to give children a positive sports experience.	KA and other community partners
Less children use screens after-school (computer, television, phone).	Organize more after-school sports activities and events to stimulate children to play outside (school 1, 2).	Promotion of active games. During after-school activities, more focus was placed on active games that children themselves could play without many extra materials.	KA
		After-school sports activities. Focused on activities that children themselves could play without too many extra materials.	KA and other community partners
		Olympic sports event. Consisted of after-school sports activities followed by a sports tournament for the four schools in the community.	KA

KA: Kids Aktief.

**Table 3 ijerph-17-00625-t003:** Examples of questions asked during the interviews with professionals.

Why did you/your school/your organization become involved in the ‘Kids in Action’ project?
What were your expectations of the project?
How did you experience the collaboration with the research team?
What do you do think we should change if we start a similar project again?
What is your opinion about the actions that were developed in collaboration with children?
How do you see the actions being taken forward after ‘Kids in Action’?
To what extent do you believe that the actions reached the children who most needed to improve their physical activity and dietary behavior?
In your opinion, how did the actions affect children’s physical activity levels and dietary behavior?
Do you believe that our project contributed to children’s participation in the community?
What development did you observe in children who participated in the Action Teams or Youth Council?
What development did you expect in children who participated in the Action Teams or Youth Council, but did not observe?
To what extent do children participate in decision making in your school/organization? - Did this change in response to ‘Kids in Action’? - How do you see the role of children in decision making for the coming years?
To what extent is children’s healthy lifestyle on the agenda of your school/organization? - Did this change in response to ‘Kids in Action’? - How does your agenda look for the coming years, in relation to children’s healthy lifestyle?

**Table 4 ijerph-17-00625-t004:** Codes used for data related to empowerment [3,23].

Categories	Codes (Times Used)
Individual	Having control (9)
	Self-efficacy (62)
	Positive self-image (46)
	Learning (67)
	Critical awareness (42)
	Critical thinking (177)
	Taking action (62)
	Participation (87)
Community	Opportunities and resources (3)
	Involvement in decision making (43)
	Value of children(‘s opinion) in community (35)
	The community wants to improve itself (0)
	Collaboration within community (19)
School	Value of children(‘s opinion) in school (14)
	Role of the school in the community (1)
	School culture (17)
	Influence of children in school decision making (28)
	Involvement of school in community decision making (1)

**Table 5 ijerph-17-00625-t005:** Number of children in the focus groups.

YPAR Groups (Abbreviation in Text)	Session Number	‘17–‘18 Begin N (Participated in Action Team Previous Year)	‘17–‘18 End N (Non-Action Team Members)	‘18–‘19 Begin N (Participated in Action Team Previous Year)	‘18–‘19 End N (Non-Youth Council Members)
Action Team 1 (AT1)	1	3 (1)	3 (0)	N.A.	N.A.
	2	4 (2)	6 (3)	N.A.	N.A.
Action Team 2 (AT2)	1	6 (4)	4 (0)	N.A.	N.A.
	2	N.A.	6 (3)	N.A.	N.A.
Action Team 3 (AT3)	1	5 (0)	4 (0)	N.A.	N.A.
	2	6 (0)	5 (0)	N.A.	N.A.
Youth Council (YC)	1	N.A.	N.A.	13 (0)	8 (0)
	2	N.A.	N.A.	13 (0)	N.A.

N.A.: not applicable.

**Table 6 ijerph-17-00625-t006:** Occupation and year since involvement of community partners interviewed.

Occupation (ID)	Actively Involved with ‘Kids in Action’ since:
School principal (S1)	Year 1
School principal (S2)	Year 1
School teacher (S3)	Year 1
School principal (S4)	Year 2
School teacher (S5)	Year 2
Policy worker at local government (G6)	Year 1
Policy worker at local government (G7)	Year 1
Social worker at community organization working with youth (C8)	Year 3
Social worker at community organization working with youth (C9)	Year 3
Social worker at community organization working with youth (C10)	Year 3
Social worker at community organization working with youth (C11)	Year 3

**Table 7 ijerph-17-00625-t007:** Codes used for data related to RE-AIM and the YPAR process.

**Theme (times used)**	**Code (Times Used)**	**Subcode (Times Used)**
	**(Related Theme in Results) ^1^**	
Community	Activities (8) (G)	
	Parental involvement (8) (G)	
	Home situation (6) (G)	
	Safety (3) (G)	
	Healthy behavior becoming the norm in the community (8) (E)(M)	
Developed activities	Opinion of children (27) (I)	
	Opinion of community partners (24) (I)	
	Not implemented (6) (I)	
	Knowledge about (5) (I)	
Adoption	Adoption of project within school (24) (A)	
	Involvement (14) (A)	Project application (5)
	Busy schedule school (10) (A)	
	Being up-to-date (63) (A)	
Most valued items	Participation of children in decision making (9) (M)	
	Organized activities (8) (M)	
	Communication (8) (M)	
	Focus on physical activity and dietary behavior (6) (M)	
	Collaboration (5) (M)	
	Youth Council (1) (M)	
	Action Teams (1) (M)	
	Children growing (1) (M)	
Reach	Motivate physically inactive children (4) (R)	
	Number of children reached (27) (R)	
Atmosphere during interview (16)		
Reasons for not reaching optimal effects	Short-term thinking (2) (E)	
	Not delivering (5) (E)	
	Focus on sustainability was too late (4) (E)	
	Rotation in staff (2) (E)	
Healthy behavior	Focus on healthy behavior at school (17) (C)	
	Habit (9) (G) (I)	
	Knowledge (4) (G)	
	Influence of weather (2) (G)	
Working together with the children	Action Teams (2) (I)	
	Other collaborations with children (10) (G)	
	Youth Council	Known in the community (7) (I)
		Busy schedule (3) (I)
		Initiation (2) (C)
		Process of meetings (13) (I)
		Composition (14) (R)
		Community partners (5) (I)
		Continuation (25) (C)
	Opinion community partners on participation (20) (I) (C) (B)	
Promotion	The study being known in the community (2) (C)	
	Recruitment (5) (C)	
Expectations (1)	Fun activity for children (2) (B)	
	None (21) (B)	
	Low (1) (B)	
	High (4) (B)	
Reason for joining the project	Similar goals (8) (B)	
	Matches or adds to own way of working (25) (B)	
	Don’t know (1) (B)	
	Role within organization (12) (B)	
	Community effort (7) (B)	
Role	Local government (10) (C) (B)	
	M.A. (20) (C) (B)	
	Partners (19) (C) (B)	
Collaboration	Flexibility (4) (C)	
	Competition (5) (C)	
	Kids Aktief (10) (C)	
	First contact (3) (B)	
	Regular contact (9) (C) (C)	
	Between organizations (14) (C)	
	Communication (23) (C)	Fast (4)
		Open (4)
	Same person of contact (1)	
Factors contributing to success (2)	Reaching many children (2) (R)	
	Being in the community (4) (E)	
	Duration of project (1) (E)	
	Children learn from participation (1) (E)	
	Children enthuse each other (9) (E)	
	No barriers to participate (2) (E) (R)	
	Listening to children (3) (E)	
	Role principal researcher (3) (E) (C)	
	Collaboration between organizations (6) (C) (E)	
	Collaboration with schools (3) (C) (E)	
	Time-investment (2) (E)	
	Trust of parents (2) (E)	
	Developed activities (13) (E) (I)	
	Children like participating (7) (E) (I)	
Sustainability (13)	Importance (11) (M)	
	Contributing to (2) (M)	
	Uncertain (13) (M)	Financial (10)
		Keep the focus (4)
		Keep the focus at school (8)
	Participation of children (10) (M) (I)	
	Role partners (19) (C) (M)	
	School continues with activities (2) (A) (M)	
	Keep improving (2) (M)	
	Hope (4) (M)	
Outcomes (8)	Applications for alternative finances for sport membership (1) (E)	
	Expectations (2) (B)	Participation in activities (4)
	Activities (3) (E)	
	Policy (3) (E)	
	Healthy lifestyle (E)	Physical activity (9)
		Dietary behavior (5)
		Attention for healthy lifestyle (8)
		Awareness physical activity (3)
		Awareness children (6)
		Awareness community partners (6)
		Temporary (6)
		Overweight (1)
	New knowledge community partners (8) (E)	
	Participation children (10) (E)	
	Empowerment children (see Table 3) (726 (E)	
	Collaboration with schools (1) (C) (E)	
Points of improvement (2)	Involve the school (4) (C)	
	Making connections in the community (6) (I)	
	Process Youth Council (3) (I)	
	Involvement of parents (9) (C)	
	Activities (2) (I)	
	Communication (4) (C)	Intermediate sharing of results (3)
		Sharing expertise (1)
		Intermediate evaluations (7)
	Person of contact (3) (C)	
	Visibility (5) (C) (I)	
	Time till action (6) (I)	

^1^ Related themes in results: A: Adoption; C: Collaboration; E: Effectiveness; G: General; I: Implementation; M: Maintenance; P: Becoming part of the project; R: Reach.

**Table 8 ijerph-17-00625-t008:** Attributes of ‘Kids in Action’ that community partners valued most.

Attributes (Times Mentioned)	Quotes
Participation of children in decision making (9)	“Participation of children in decision making because we have seen that if it [actions] comes from them then it works better.” [S3] “Participation of children and the Action Teams for me is the same, because participation of children is the Action Team.” [C11] “Child participation is very important. Look, with everything that we do, it’s very important. Because they feel, when you let children participate, not only in the Youth Council, but also in our organization, they feel ‘*It’s mine’. I have something, I have achieved something. Look, it’s not only for me, it’s for everybody. We have achieved something*.’ I see that in their behavior. I think that’s very good.” [C8]
Organized actions (8)	“Organized activities, especially the maintenance of it I think is important.”[S1]
Communication (8)	“Communication is always the most important, especially when it concerns schools.” [S2] “Collaboration and communication is very important because that’s actually where it all starts.” [C11] “And maybe a point of improvement is yes the research results. That in between you can already share some of them.” [G6]
Focus on physical activity and dietary behavior (6)	“Healthy behavior because it has become an important focus of the Youth Council this year.” [C10] “I think that this was the main goal [healthy behavior]. I also think that for our school, to see if we can influence this... I really think that this was achieved and that at school this was also achieved and that our team really became aware of that. The school management thinks this is important and it’s something that we as a school want to represent. For us, this was very important.” [S5]
Collaboration (5)	“Collaboration because the collaboration with you was very good. Having all your support, it really helped. And, how to handle the kids, yes the collaboration was great. Actually I don’t want to lose you. [laughs]” [C8] “The time and energy you have invested in it has been very important.” [G6] “The collaboration between stakeholders and then I actually think about the collaboration between our school and you, I always experienced that as very pleasant. Very clear.” [S5]
Youth Council (1)	“The children finally have the right to speak up about what they want in their community and what they want to do about it.” [C8]
Action Teams (1)	“As a teacher I see a lot of positive results from this. I think that this, for children to be a part of it and get the opportunity to think along, that’s very important for their development. You know, that their voice matters and that they learn how to reach other children with that, how they should deliver a message. That’s not only beneficial here at school, but also for the rest of their lives it’s very important that they learn a lot of things… skills.” [S5] “The children in the Action Teams, well that was very positive. And yes very, I think also for the community very important. That something like that is there and that something like that continues to exist. It’s necessary and it connects people.” [S5]
Children growing (1)	“Growth in children is very important. I see them grow every day. The children of the Youth Council grow a lot. With what they have reached and what they still want to achieve.” [C8]

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
