# Peer review of "“Not Only Adults Can Make Good Decisions, We as Children Can Do That as Well” Evaluating the Process of the Youth-Led Participatory Action Research ‘Kids in Action’"

_ijerph, 2020, doi:10.3390/ijerph17020625_

Round 1

Reviewer 1 Report

Manuscript Title: “Not only adults can make good decisions, we as children can do that as well:” Evaluating the process of the youth-led participatory action research ‘Kids in Action’.

________________________________________________________ 

This article provided the results of a process evaluation of the youth-led participatory action research project “Kids in Action.” This project was implemented over three years in partnership with 9-12 year olds; two years in schools and one year in the community in the form of a youth council. The authors utilized focus groups and interviews for the process evaluation using an empowerment and RE-AIM framework. Youth, school staff, and policy makers participated. The authors reported positive changes in youth and community empowerment; organizational was more challenging.   

Main Strengths/Weaknesses: An overall strength of this submission was that the authors utilized guiding theoretical frameworks. Often, even if theoretical frameworks are used in the YPAR literature, they are not mentioned in papers. It was also a strength that they assessed change at multiple levels, and from multiple stakeholder perspectives.  They included up-to-date references related to YPAR/youth CBPR, including recent systematic reviews.  A strength of the evaluation itself was that youth could bring friends from school to participate in the evaluation – this is a great way to assess reach to students that are not a part of the core team, but should still be impacted by the actions. Finally, the authors are filling the gap in the YPAR literature related to conducting and reporting the results of a process evaluation.  

A major weakness was that specific information about YPAR implementation itself was not assessed/provided. It is helpful to understand stakeholder perspectives related to implementation during process evaluations like the authors obtained, but just as important are reflections from the implementers themselves about what did/did not go as planned, mid-course corrections, and proposed changes for future implementation. Including this perspective will also be helpful for readers, as it will help them in planning their own projects more so than reading qualitative descriptions of stakeholders’ accounts related to changes in outcomes.

A second major area for improvement was the bigger picture; information on the overall structure of the larger project. The authors do reference other manuscripts, but more information about the other pieces of the work, such as the actions from these projects, will be helpful for readers to understand the project in context. For example, near the end of the paper, the reader discovers that youth were not involved in forming the project from the beginning, and did not decide on the topic area (physical activity and nutrition). It would be fruitful to state that at the end of the introduction/beginning of the method. Since youth did not decide, it will be important to include who did, and why the partnership formed between the schools, community groups, and  researchers. It would also be helpful for the authors to note that youth participatory action research is conceptualized on a continuum – from gaining input from youth to youth leadership without adult involvement.

Suggestions for Overall Improvement:  To improve the usefulness of the process evaluation findings for other researchers/community practitioners, it will be helpful for the authors to provide information on the YPAR process, its strengths and challenges, and any course corrections. For example, how easy was it for youth to learn research methods? Did they only choose certain methods with this age range? Was the YPAR process modified to fit the youths’ developmental level? How did youth advocate for the actions and would they do anything differently? Including answers to questions such as these will make this paper more useful to readers than it is in its current form.

Related to the bigger picture of the work, more information on the partnership formation should be provided, and why the study came about. To place the study in the larger framework of youth participation, the authors should comment in the introduction or method about where this study falls on the participation continuum. Using the framework proposed in the article by Jacquez, Vaughn and Wagner (2013)  that they cited would be  a helpful way of framing it – listing the phases of research in which youth were involved.

Specific Suggestions:

Introduction:

Even though the actions were reported in the outcomes paper, for readers to understand this project, it will be important to include youth actions in this paper too. Listing them in a table by year and school/council would work well. Understanding the timeline of the project in the schools and community group will give readers a better understanding of this work. It will be helpful to include a timeline for each year. For example, when did youth learn research methods, gather data, and advocate for their actions?

Method:

Please include any inclusion/exclusion criteria. For example, how did teachers select students to participate? What are the strengths and challenges of using these strategies?

There are some phrases in the paper that could use clarifying for readers. For example, the authors reference a participatory needs assessment and include a citation to a previous paper. It would be useful to include at least one sentence about what was found in that participatory needs assessment in this paper.  On page two, the authors should clarify “ideas of children and behavior change theories” and how they were combined using intervention mapping methodology. In this instance, a citation is provided, but a sentence describing what this means is important for the reader. I would also re-phrase the phrase in quotations above, as it is difficult to understand what the authors mean. On page three, re-word the sentence “The Action Teams came up with ideas how to reach the goals, leading to ideas for actions” as it is vague. On page three, elaborate more on how the following promotes continuation:  “In order to promote continuation of the YPAR process beyond the research project, instead of one ActionTeam per school, in the third year one Action Team was started with representatives of three schools, called the Youth Council.” In the following statement: “We used literature on empowerment theory to develop the focus groups with children.” Does this mean that empowerment theory was used to inform the focus group questions and protocol? Related to the questions and protocol, a table with the interview and focus group questions, or at least examples of questions, would be helpful for readers to understand the process evaluation further. On page four, what is meant by “assignments?” Please clarify. The authors state that no actions were developed in year one. If the youth did not focus on actions, what did they do? Again, this brings it back to providing more information on the implementation of YPAR itself. It will be helpful to include reasons why one school no longer participated.

Analyses & Results:

Please include citations for the coding processes magnitude, descriptive, and emotion coding. The results could be organized in a way that is more meaningful for readers. For example, the quotes in the table were great. However, some parts of the paragraphs felt like general, vague lists that anyone could generate. It will be helpful to include stakeholder quotes within those lists, or make a general statement about the finding, with a quote to illustrate. That is already done some, but results could use more structure like that.

Discussion:

Additional limitations of the work were that youth weren’t involved from the beginning, adults formed the groups, and the project started at the organizational level versus youth level. The authors do report these in the discussion, but moving them to the limitations would be useful. A limitation to add is that the youth did not conduct the process evaluation. If they did participate in conducting it, then that should be made clear. Finally, it would be helpful for the authors to potentially cite these two recent papers  on YPAR and systems change and YPAR in schools:

Kennedy, H., DeChants, J., Bender, K., & Anyon, Y. (2019). More than Data Collectors: A Systematic Review of the Environmental Outcomes of Youth Inquiry Approaches in the United States. American journal of community psychology63(1-2), 208-226.

Anderson, A. J. (2019). A Qualitative Systematic Review of Youth Participatory Action Research Implementation in US High Schools. American journal of community psychology.

Overall, this paper can contribute meaningfully to the literature if information about YPAR implementation is provided, including strengths and challenges, adaptations, and course corrections, and if the authors provide more nuanced findings. The authors should also note where their project falls on the participatory continuum and provide more information about the actions, partnership, and participatory needs assessment. The paper does have strengths of connections to theory, filling a gap in the literature related to process evaluation, and evaluating change at multiple levels from multiple perspectives.

Author Response

This article provided the results of a process evaluation of the youth-led participatory action research project “Kids in Action.” This project was implemented over three years in partnership with 9-12 year olds; two years in schools and one year in the community in the form of a youth council. The authors utilized focus groups and interviews for the process evaluation using an empowerment and RE-AIM framework. Youth, school staff, and policy makers participated. The authors reported positive changes in youth and community empowerment; organizational was more challenging.   

Main Strengths/Weaknesses: An overall strength of this submission was that the authors utilized guiding theoretical frameworks. Often, even if theoretical frameworks are used in the YPAR literature, they are not mentioned in papers. It was also a strength that they assessed change at multiple levels, and from multiple stakeholder perspectives. They included up-to-date references related to YPAR/youth CBPR, including recent systematic reviews.  A strength of the evaluation itself was that youth could bring friends from school to participate in the evaluation – this is a great way to assess reach to students that are not a part of the core team, but should still be impacted by the actions. Finally, the authors are filling the gap in the YPAR literature related to conducting and reporting the results of a process evaluation.  

Our reply: We would like to thank the reviewer for taking the time to review our paper and for acknowledging the strengths and added value of our study. Hereafter, we reply to the specific comments of the reviewer in blue and have highlighted the changes in our manuscript.

A major weakness was that specific information about YPAR implementation itself was not assessed/provided. It is helpful to understand stakeholder perspectives related to implementation during process evaluations like the authors obtained, but just as important are reflections from the implementers themselves about what did/did not go as planned, mid-course corrections, and proposed changes for future implementation. Including this perspective will also be helpful for readers, as it will help them in planning their own projects more so than reading qualitative descriptions of stakeholders’ accounts related to changes in outcomes.

Our reply: we thank the reviewer for this critical evaluation. We agree with the reviewer that specific information about YPAR implementation itself is important. We have provided a detailed description about the YPAR process as it evolved including an evaluation of this process elsewhere (Anselma et al., 2019). That paper also includes the experiences of the implementers/authors. The current paper is a retrospective evaluation of the professionals involved in the study and an evaluation of the participatory process and actions with the children who actively participated in the study. We specified this in the text, as we would like to keep the distinction between this paper and Anselma et al., 2019, preventing duplications. To clarify this we added the following (lines 72-73): “For a detailed description of the methodological process of combining YPAR with IM and how this was experienced by the researchers, we refer to Anselma et al., 2019b[19]”.
We have further specified that it was an iterative process, so there was no evaluation of the planning or mid-course corrections (lines 90-96): “YPAR was combined with Intervention Mapping (IM), to structure the process of action development and relate children’s ideas to evidence-based behavior change strategies[19-22]. Children were involved throughout the IM-process as much as possible, with certain IM-steps being adapted to be suitable for children. Some theoretical tasks were performed by an IM expert panel as they required specific knowledge that would be difficult or too time-consuming to teach the children. The process of combining IM and YPAR was iterative and is described in detail in Anselma et al., 2019b[19].”

A second major area for improvement was the bigger picture; information on the overall structure of the larger project. The authors do reference other manuscripts, but more information about the other pieces of the work, such as the actions from these projects, will be helpful for readers to understand the project in context. For example, near the end of the paper, the reader discovers that youth were not involved in forming the project from the beginning, and did not decide on the topic area (physical activity and nutrition). It would be fruitful to state that at the end of the introduction/beginning of the method. Since youth did not decide, it will be important to include who did, and why the partnership formed between the schools, community groups, and  researchers.

Our reply: we thank the reviewer for this helpful suggestion. We have added more information about the overall structure on the project in section 2.1. Study outline (lines 76-96) and included Table 2, which presents the developed actions on page 5-6.

It would also be helpful for the authors to note that youth participatory action research is conceptualized on a continuum – from gaining input from youth to youth leadership without adult involvement.

Our reply: the reviewer makes a great point and we have added a paragraph in the introduction on youth participation and where on the participation ladder we believe our research took place (lines 54-59): Throughout the study we tried to collaborate with children on the level of shared decision making – i.e. level 6 of Hart’s children’s participation ladder[14]. The overall research aim of improving children’s lifestyle was decided upon by researchers, but during the rest of the study children were actively involved as partners, from actively giving input through participatory meetings, conducting research, analyzing results and implementing and evaluating actions[9].”

Suggestions for Overall Improvement:  To improve the usefulness of the process evaluation findings for other researchers/community practitioners, it will be helpful for the authors to provide information on the YPAR process, its strengths and challenges, and any course corrections. For example, how easy was it for youth to learn research methods? Did they only choose certain methods with this age range? Was the YPAR process modified to fit the youths’ developmental level? How did youth advocate for the actions and would they do anything differently? Including answers to questions such as these will make this paper more useful to readers than it is in its current form.

Our reply: we thank the reviewer for these suggestions. However these questions are included in another paper as mentioned previously (Anselma et al., 2019). We did not evaluate these questions for the present paper. To clarify this we have added a sentence in the introduction to refer to this previous paper (lines 72-73): “For a detailed description of the methodological process of combining YPAR with IM and how this was experienced by the researchers, we refer to Anselma et al., 2019b[19].”

Related to the bigger picture of the work, more information on the partnership formation should be provided, and why the study came about.

Our reply: we have provided more information about the collaborations and how they came about (lines 76-82): “Researchers from the Amsterdam UMC collaborated with the sports-based daycare ‘Kids Aktief’ (KA) from the conception and application for funding. Together with the local government it was decided to focus on one specific community in their district that would benefit most from a YPAR study aimed at stimulating a healthy lifestyle of children. When setting up the study, the policy makers invited the research team to join an existing community project group on healthy lifestyle of children in the community. Textbox A describes the most important participants and community partners of this study.”

To place the study in the larger framework of youth participation, the authors should comment in the introduction or method about where this study falls on the participation continuum. Using the framework proposed in the article by Jacquez, Vaughn and Wagner (2013)  that they cited would be  a helpful way of framing it – listing the phases of research in which youth were involved.

Our reply: we thank the reviewer for this valuable suggestion and we have added a paragraph in the introduction on youth participation and where on the participation ladder we believe our research took place (lines 54-66): “Throughout the study we tried to collaborate with children on the level of shared decision making – i.e. level 6 of Hart’s children’s participation ladder[14]. The overall research aim of improving children’s lifestyle was decided upon by researchers, but during the rest of the study children were actively involved as partners, from actively giving input through participatory meetings, conducting research, analyzing results and implementing and evaluating actions[9]. Children became partners starting with the participatory needs assessment, where children, parents and community professionals identified two main needs that specified the aim of the current study (i.e. improve physical activity and a healthy diet). To gain a deeper understanding of the more and less effective elements of actions, as well as facilitators and barriers for sustainable implementation[15.16], academic researchers conducted an extensive process evaluation. Children were not involved as partners in the process evaluation because it was conducted over several years and we worked with different groups of children each year. Moreover, the extensive process evaluation would take too much time, which children needed to develop actions.”

Specific Suggestions:

Introduction:

Even though the actions were reported in the outcomes paper, for readers to understand this project, it will be important to include youth actions in this paper too. Listing them in a table by year and school/council would work well.

Our reply: we thank the reviewer for this suggestion and have inserted Table 2 with the developed actions on page 5-6.

Understanding the timeline of the project in the schools and community group will give readers a better understanding of this work. It will be helpful to include a timeline for each year. For example, when did youth learn research methods, gather data, and advocate for their actions?

Our reply: We understand that a timeline could be helpful. However, the timeline varied between schools and could also be very different for future studies as it is very dependent on the schools, participating children, when in the school year the project starts, topic and other community characteristics. Therefore, we decided against adding a study timeline. In the paper we refer to (Anselma et al., 2019b) we offer a step-by-step explanation of intervention mapping and its application in the Kids in Action study, which can assist readers in their understanding of the timeline.

Method:

Please include any inclusion/exclusion criteria. For example, how did teachers select students to participate? What are the strengths and challenges of using these strategies?

Our reply: we have added more details on the selection process (lines 119-122): “Selection procedures varied per school and per year. For example, in the first year, at one school all children could sign up; at two schools the meetings were held during school hours, therefore teachers selected children who could miss academic time; at another school, children who were part of the student council were invited to participate in the Action Team.”

There are some phrases in the paper that could use clarifying for readers. For example, the authors reference a participatory needs assessment and include a citation to a previous paper. It would be useful to include at least one sentence about what was found in that participatory needs assessment in this paper.  

Our reply: we apologize that some phrases were not clear. We have added more information about the participatory needs assessment (lines 82-87): “The ‘Kids in Action’ study consisted of two phases, of which phase 1 started in 2015 with a participatory needs assessment. Research was conducted in collaboration with children and interviews were held with parents and professionals from the community project group. Results showed that low levels of physical activity and unhealthy dietary behavior were perceived as the main health problems children in the community faced. Therefore, improving these behaviors in 9-12-year old children became the focus of phase 2, which started in 2016 and lasted for 3 years[20, 21].

On page two, the authors should clarify “ideas of children and behavior change theories” and how they were combined using intervention mapping methodology. In this instance, a citation is provided, but a sentence describing what this means is important for the reader.

Our reply: we apologize that some phrases were not clear. We have rewritten the sentence and added more information about the paper we refer to (lines 88-96): “The current process evaluation focuses on phase 2, where children were involved throughout the process of action development, implementation and evaluation, from doing background research to developing and evaluating actions. YPAR was combined with Intervention Mapping (IM), a stepwise approach to developed evidence-based actions, to structure the process of action development and relate children’s ideas to evidence-based behavior change strategies[19, 22]. Children were involved throughout the IM-process as much as possible, with certain IM-steps being adapted to be suitable for children. Some theoretical tasks were performed by an IM expert panel as they required specific knowledge that would be difficult or too time-consuming to teach the children. The process of combining IM and YPAR was iterative and is described in detail in Anselma et al., 2019b[19].

I would also re-phrase the phrase in quotations above, as it is difficult to understand what the authors mean.

Our reply: we apologize that this phrase was not clear. We have reworded it, as follows (lines 90-92): “YPAR was combined with Intervention Mapping (IM), a stepwise approach to developed evidence-based actions, to structure the process of action development and relate children’s ideas to evidence-based behavior change strategies[19, 22].”

On page three, re-word the sentence “The Action Teams came up with ideas how to reach the goals, leading to ideas for actions” as it is vague.

Our reply: we apologize that this sentence was not clear. We have reworded it as follows (lines 128-130): The Action Teams thought of ways to reach the goals, voted for the best ideas and then further specified the ideas by making production and implementation plans.”

On page three, elaborate more on how the following promotes continuation:  “In order to promote continuation of the YPAR process beyond the research project, instead of one Action Team per school, in the third year one Action Team was started with representatives of three schools, called the Youth Council.”

Our reply: we have added the following to explain how the Youth Council aided continuation of child participation in the community (lines 139-145): The principal researcher (M.A.) hosted and facilitated the Youth Council together with representatives of three community organizations and trained them in facilitating child participation. One social worker of one of the community organizations co-facilitated the Youth Council with the principal researcher throughout the year, with the goal to host the Youth Council by herself with a new assistant the following years. This collaboration between M.A. and the three community organizations ensured continuation of child participation in policy of community partners after the research project.”

In the following statement: “We used literature on empowerment theory to develop the focus groups with children.” Does this mean that empowerment theory was used to inform the focus group questions and protocol?

Related to the questions and protocol, a table with the interview and focus group questions, or at least examples of questions, would be helpful for readers to understand the process evaluation further.

Our reply: we indeed used empowerment theory to inform the focus group questions and protocol. We have reworded the sentence to clarify this and moved it to a different paragraph (lines 167-168): “We used literature on empowerment theory to develop the protocol for the focus groups with children[3, 23].”. We also included Table 3 with examples of questions from the interview guide (page 8).

On page four, what is meant by “assignments?” Please clarify.

Our reply: the focus groups consisted of two assignments which we explained in detail in Anselma et al., 2019a and b. We have included a general explanation on lines 168-186.

The authors state that no actions were developed in year one. If the youth did not focus on actions, what did they do? Again, this brings it back to providing more information on the implementation of YPAR itself. It will be helpful to include reasons why one school no longer participated.

Our reply: we apologize that this was not clear. We have rewritten the sentence to clarify what we focused on in the first year (lines 163-165): “No actions were developed in year one because the first year was mainly dedicated to completing the needs assessment. Therefore, the first focus groups took place at the beginning of year two.
We also provided a reason for the school dropping out (lines 132-133): “One school decided not to participate in the second year because they chose to participate in another study. They decided to re-join the project in the third year.

Analyses & Results:

Please include citations for the coding processes magnitude, descriptive, and emotion coding.

Our reply: we apologize that this citation was not placed at the correct position, we adjusted this (lines 217-221): “We used Evaluation Coding as a basis to code topics other than empowerment, and combined this with several other coding methods such as Magnitude Coding, Descriptive Coding and Emotions Coding, to be able to differentiate between positive and negative items, to label emotions and experiences, and to be able to give value to the variety of materials[25].

The results could be organized in a way that is more meaningful for readers. For example, the quotes in the table were great. However, some parts of the paragraphs felt like general, vague lists that anyone could generate. It will be helpful to include stakeholder quotes within those lists, or make a general statement about the finding, with a quote to illustrate. That is already done some, but results could use more structure like that.

Our reply: we thank the reviewer for this helpful suggestion. We have added more quotes to illustrate our findings. Moreover, we have restructured certain parts, as in the following example (lines 513-523): “Some points for improvement were suggested in the process of collaborating with children. First, children participating in the Action Teams and Youth Council could involve more children in the decision making process, making it a project of all children instead of only the ones in the participatory groups. Second, it was considered a shame that one school did not participate in the Youth Council, due to extracurricular activities happening at that school at the same time as the Youth Council’s meetings[S4, C10]. For the following year the schedule of the Youth Council should be better suited to all schools. Third, the low parental involvement was mentioned as a point of improvement, but was acknowledged as a challenge and time investment[S3, S5, G7, C10, C11]: “Maybe we could have invested more in parent participation. I think that should also be a focus point for the Youth Council for next year. With the cooking classes we managed to have some parents present, but it is very difficult and you have to be lucky to find a mother who is interested. But it is definitely something we should focus on.”[C11].”

Discussion:

Additional limitations of the work were that youth weren’t involved from the beginning, adults formed the groups, and the project started at the organizational level versus youth level. The authors do report these in the discussion, but moving them to the limitations would be useful.

Our reply: this is a helpful suggestion and we have moved the section to the limitations section (lines 677-681): “Also, we did not include children in the initial design during the grant application phase. This could have benefited mutual understanding between researchers and children from the beginning, stating boundaries and discussing where concessions can be made[7, 38]. However, applying for funding is a lengthy and uncertain process. Recruiting schools for a project planned in future school years that might not even get funded is challenging.”

A limitation to add is that the youth did not conduct the process evaluation. If they did participate in conducting it, then that should be made clear.

Our reply: we acknowledge this limitation and have added it to the limitations section (lines 681-685): “Furthermore, we did not conduct the process evaluation with children as partners, because it did not fit the timeline of the study and would be an additional task and time investment for the children. Future YPAR studies may explore ways to evaluate the entire research process together with children as partners. This may provide additional relevant insights to strengthen YPAR studies.”

Finally, it would be helpful for the authors to potentially cite these two recent papers  on YPAR and systems change and YPAR in schools:

Kennedy, H., DeChants, J., Bender, K., & Anyon, Y. (2019). More than Data Collectors: A Systematic Review of the Environmental Outcomes of Youth Inquiry Approaches in the United States. American journal of community psychology63(1-2), 208-226.

Anderson, A. J. (2019). A Qualitative Systematic Review of Youth Participatory Action Research Implementation in US High Schools. American journal of community psychology.

Our reply: we thank the reviewer for pointing out these interesting reviews and we have incorporated them in our paper:
YPAR has mostly been applied in school-settings to address problems with education, social inequalities, health, or the physical and social environment[10, 11].” (lines 43-45)

“A recent review found that through being involved in YPAR, adults appreciated children’s abilities and valued their participation[11].” (lines 595-596)

“A recent review also found that engaging youth in inquiry aided the research process, for example because youth can stimulate each other to participate in actions[11].” (lines 607-609)

When children are involved in advocacy and organizing, this can lead to environmental outcomes such as changes in peer norms and program development[11].” (lines 612-614)

“Collaborations between academics and community organizations may provide an enabling environment for successful YPAR[34].” (lines 654-655)

“It also required high flexibility from researchers and the community project group, as YPAR remains an iterative approach dependent on many uncontrollable factors and school schedules[34, 35].” (lines 672-674)

Overall, this paper can contribute meaningfully to the literature if information about YPAR implementation is provided, including strengths and challenges, adaptations, and course corrections, and if the authors provide more nuanced findings. The authors should also note where their project falls on the participatory continuum and provide more information about the actions, partnership, and participatory needs assessment. The paper does have strengths of connections to theory, filling a gap in the literature related to process evaluation, and evaluating change at multiple levels from multiple perspectives.

Our reply: we thank the reviewer for taking the time to review our paper. We have revised the paper based on these suggestions and hope the revisions are clear and satisfactory.

Reviewer 2 Report

This is a valuable paper, as it presents a rare “process evaluation” of an important body of participatory action research. It is well-written, to a suitable professional standard for journal publication with only minor revision needed. The following suggestions are offered in the spirit of supportive constructive critique, and in the hope that they will be found helpful…

“Youth”

The English word “youth” describes a condition of youthfulness. It is often also used as a term to describe a “young person”, or the class of young people in general, but this usage now seems dated, as it tends to label the young person as a member of a generic social classification rather than as a real person. Also I note that the participants in this action research project were aged 9-12, so “children” would be a more appropriate description in this context. I suggest, where you are referring to people under 13, use “child” or “children”, and only where you are explicitly referring to 13-and-overs, use “adolescent” or “young person”. You should also change the sub-title for consistency; this is clearly “child-led PAR”, not “Youth-led”.

There is a problem here, in that you have drawn on a methodological literature and a methodological approach which you call “Youth-led Participatory Action Research (YPAR)”, so I would suggest adding a brief discussion on why you decided to draw on a “youth-led” rather than a “child-led” methodological approach when you are doing action research with children, not with adolescents.

Invitation to join an adult-defined enterprise.

I think it is important to note somewhere that your approach involved inviting children to join a project whose thematic area and health-related aims had already been decided on by adults as being of interest and value to them. The children were not asked to decide for themselves what problems they faced in their lives and in their communities that needed researching. (By the way, most adult researchers spend their working lives researching problems that have been defined by others, so it does not devalue this research, but I think it should be addressed).

Translation

It appears that the research was carried out in and around Amsterdam. If so, was the data collected in Dutch? Was the analysis carried out on the original data in Dutch? The paper contains extensive quotes in English, so if the data was in Dutch, there must have been translation at some point. Who did it? All this is part of your research methodology so should be made visible in the report.

Quote selection

The extensive quotes from participants are a valuable aspect of this report. The quotes shown, however, represent only a tiny fraction of the data collected. Biased selection of quotes in a report is one of the easiest ways for researchers to manipulate the meaning of their findings, so to avoid this, the method used for selecting which quotes to put in the report needs to be clearly visible. Normally it is best to select quotes that are representative of the data in general, and then when you use a quote that represents a differing or incongruent point to view, make explicit mention of this as you go along.

Adult power over epistemology

I was struck by the sentence, “To keep the integrity and methodological quality of the research high, researchers need to control research methodology and align actions with health behavior theories”. This is too simplistic and hides a lot of assumptions about the use of adult power over children. It is about adults using their power to make sure children do what the adults think is right, as opposed to adults supporting children to do what the children think is right. When you say, “sharing power is not necessarily needed for every decision”, the sub-text is, “sometimes sharing power is best avoided to make sure adults retain control”. Again in the discussion about cooking, the choice of words is significant. Where you say “children wanted a combination of healthy and unhealthy foods”, I wonder if the truth was that the children wanted to cook tasty food they actually enjoyed eating, and the adults felt duty-bound to prevent this? Of course, I wasn’t there, so I can’t say for sure. Given the context you were working in, I don’t want to be over-critical, but I feel this deserves a more sophisticated discussion.

Author Response

Comments and Suggestions for Authors

This is a valuable paper, as it presents a rare “process evaluation” of an important body of participatory action research. It is well-written, to a suitable professional standard for journal publication with only minor revision needed. The following suggestions are offered in the spirit of supportive constructive critique, and in the hope that they will be found helpful…

Our reply: We would like to thank the reviewer for the taking the time to review our paper and for the very helpful comments. Hereafter, we reply to the specific comments of the reviewer in blue and have highlighted the changes in our manuscript.

“Youth”

The English word “youth” describes a condition of youthfulness. It is often also used as a term to describe a “young person”, or the class of young people in general, but this usage now seems dated, as it tends to label the young person as a member of a generic social classification rather than as a real person. Also I note that the participants in this action research project were aged 9-12, so “children” would be a more appropriate description in this context. I suggest, where you are referring to people under 13, use “child” or “children”, and only where you are explicitly referring to 13-and-overs, use “adolescent” or “young person”. You should also change the sub-title for consistency; this is clearly “child-led PAR”, not “Youth-led”.

There is a problem here, in that you have drawn on a methodological literature and a methodological approach which you call “Youth-led Participatory Action Research (YPAR)”, so I would suggest adding a brief discussion on why you decided to draw on a “youth-led” rather than a “child-led” methodological approach when you are doing action research with children, not with adolescents.

Our reply: we thank the reviewer for this valuable insight. As our study builds on YPAR literature, we use ‘youth’ when we discuss literature or methodology; when talking about our study, we switch to ‘children’. We did notice we did not do this consistently, so we changed it accordingly. We have also added a sentence in the introduction to clarify that we switch from youth to children (lines 49-52): “‘Kids in Action’ is such a YPAR study, in which academic researchers collaborate with 9-12-year-olds (further named ‘children’) from a low SEP neighborhood in order to improve their physical activity and dietary behavior.
We also understand the point the reviewer makes about using ‘child-led PAR’, instead of ‘youth-led PAR’. But as we used ‘youth-led PAR’ in our previous papers about this study, we would like to keep it as it is for consistency. But we will definitely take in into account for future projects.

Invitation to join an adult-defined enterprise.

I think it is important to note somewhere that your approach involved inviting children to join a project whose thematic area and health-related aims had already been decided on by adults as being of interest and value to them. The children were not asked to decide for themselves what problems they faced in their lives and in their communities that needed researching. (By the way, most adult researchers spend their working lives researching problems that have been defined by others, so it does not devalue this research, but I think it should be addressed).

Our reply: the thematic area of healthy behavior was chosen by the researchers, but in the participatory needs assessment, children, adults and local professionals decided on the focus of physical activity and dietary behavior. To clarify this, we have added more information about the participatory needs assessment and the process at the start of the study (lines 76-87): “Researchers from the Amsterdam UMC collaborated with the sports-based daycare ‘Kids Aktief’ (KA) from the conception and application for funding. Together with the local government it was decided to focus on one specific community in their district that would benefit most from a YPAR study aimed at stimulating a healthy lifestyles of children. When setting up the study, the policy makers invited the research team to join an existing community project group working on healthy lifestyles of children in the community. Textbox A describes the most important participants and community partners of this study. The ‘Kids in Action’ study consisted of two phases, of which phase 1 started in 2015 with a participatory needs assessment. Research was conducted in collaboration with children, and interviews were held with parents and professionals from the community project group. Results showed that low levels of physical activity and unhealthy dietary behavior were perceived as the main health problems children in the community faced. Therefore, improving these behaviors in 9-12-year old children should be the focus of phase 2, which started in 2016 and lasted for 3 years[20, 21].”

Translation

It appears that the research was carried out in and around Amsterdam. If so, was the data collected in Dutch? Was the analysis carried out on the original data in Dutch? The paper contains extensive quotes in English, so if the data was in Dutch, there must have been translation at some point. Who did it? All this is part of your research methodology so should be made visible in the report.

Our reply: indeed the data was collected and analyzed in Dutch. We apologize that we did not have information about this in the methods. We have added a sentence on this in the methods section (lines 221-222): “All analyses were conducted in Dutch and quotes were translated by M.A. for the current paper.”

Quote selection

The extensive quotes from participants are a valuable aspect of this report. The quotes shown, however, represent only a tiny fraction of the data collected. Biased selection of quotes in a report is one of the easiest ways for researchers to manipulate the meaning of their findings, so to avoid this, the method used for selecting which quotes to put in the report needs to be clearly visible. Normally it is best to select quotes that are representative of the data in general, and then when you use a quote that represents a differing or incongruent point to view, make explicit mention of this as you go along.

Our reply: we apologize that this information was missing. We first wrote the text based on the results from the analyses and then looked for quotes to support the text. We have added this to the methods section (lines 222-224): “Quotes that could serve as an example of, or explain a piece of text, were also selected and subsequently translated by M.A..”

Adult power over epistemology

I was struck by the sentence, “To keep the integrity and methodological quality of the research high, researchers need to control research methodology and align actions with health behavior theories”. This is too simplistic and hides a lot of assumptions about the use of adult power over children. It is about adults using their power to make sure children do what the adults think is right, as opposed to adults supporting children to do what the children think is right. When you say, “sharing power is not necessarily needed for every decision”, the sub-text is, “sometimes sharing power is best avoided to make sure adults retain control”. Again in the discussion about cooking, the choice of words is significant. Where you say “children wanted a combination of healthy and unhealthy foods”, I wonder if the truth was that the children wanted to cook tasty food they actually enjoyed eating, and the adults felt duty-bound to prevent this? Of course, I wasn’t there, so I can’t say for sure. Given the context you were working in, I don’t want to be over-critical, but I feel this deserves a more sophisticated discussion.

Our reply: we thank the reviewer for this critical and valuable note. We have critically looked at the phrases the reviewer points out and rewritten them. We hope that this helps to show that we did not want to hold on to adult power, but that sometimes shared power is not the way to go when either children or adults need to hold on to their beliefs and methods (lines 614-619 and 621-624): It should be kept in mind that sharing power may not be possible for every decision. To preserve scientific integrity and methodological quality, researchers may need to guarantee adherence to guidelines regarding scientific integrity and methodology and in the case of our study align the developed actions with health behavior theories[19]. On the other hand, when children feel strongly about something that researchers do not recognize, children could be allowed to go through with it. Through YPAR children and researchers educate each other[28, 29].”
“Through dialogue, children came to understand that the focus had to be on healthy foods as that was the goal of the project and children helped adults to design attractive leaflets and content of the workshops that would appeal to children, i.e. healthy meals that children enjoy.”

Also, we have added a paragraph in the introduction on youth participation and where on the participation ladder we believe our research took place. We hope that this helps to put our research and views in perspective (lines 54-66): “Throughout the study we tried to collaborate with children on the level of shared decision making – i.e. level 6 of Hart’s children’s participation ladder[14]. The overall research aim of improving children’s lifestyle was decided upon by researchers, but during the rest of the study children were actively involved as partners, from actively giving input through participatory meetings, conducting research, analyzing results and implementing and evaluating actions[9]. Children became partners starting with the participatory needs assessment, where children, parents and community professionals identified two main needs that specified the aim of the current study (i.e. improve physical activity and a healthy diet). To gain a deeper understanding of the more and less effective elements of actions, as well as facilitators and barriers for sustainable implementation[15, 16], academic researchers conducted an extensive process evaluation. Children were not involved as partners in the process evaluation because it was conducted over several years and we worked with different groups of children each year. Moreover, the extensive process evaluation would take too much time, which children needed to develop actions.”

Round 2

Reviewer 1 Report

The authors have addressed all of my suggested changes adequately, and I have no additional changes to suggest. There is only some minor text editing needed to correct grammatical errors and sentence structure. 

Author Response

We thank the reviewer for looking through our manuscript again. We have carefully read the manuscript and corrected errors. We have highlighted them in the text.